# Genome Identification and Evolutionary Analysis of *LBD* Genes and Response to Environmental Factors in *Phoebe bournei*

**DOI:** 10.3390/ijms241612581

**Published:** 2023-08-09

**Authors:** Yiming Ma, Minchen Zhong, Jingshu Li, Yiming Jiang, Xuerong Zhou, Chris Justice Ijeoma, Xinghao Tang, Shipin Chen, Shijiang Cao

**Affiliations:** 1International College, Fujian Agriculture and Forestry University, Fuzhou 350002, Chinachrisfafu@outlook.com (C.J.I.); 2College of Forestry, Fujian Agriculture and Forestry University, Fuzhou 350002, China; 3Horticultrue College, Fujian Agriculture and Forestry University, Fuzhou 350002, China; 4Commonwealth Scientific Industrial Research Organization (CSIRO) Agriculture Food, Canberra, ACT 2601, Australia; xue-rong.zhou@csiro.au; 5Fujian Academy of Forestry, Fuzhou 350002, China

**Keywords:** *LBD* gene family, *Phoebe bournei*, conserved domains, LBD transcription factor, light qualities, abiotic stress

## Abstract

*Phoebe bournei* is nationally conserved in China due to its high economic value and positive effect on the ecological environment. *P. bournei* has an excellent wood structure, making it useful for industrial and domestic applications. Despite its importance, there are only a few studies on the lateral organ boundary domain (LBD) genes in *P. bournei*. The *LBD* gene family contributes to prompting rooting in multiple plant species and therefore supports their survival directly. To understand the LBD family in *P. bournei*, we verified its characteristics in this article. By comparing the sequences of *Arabidopsis* and identifying conserved domains and motifs, we found that there were 38 members of the LBD family in *P. bournei*, which were named *PbLBD1* to *PbLBD38*. Through evolutionary analysis, we found that they were divided into two different populations and five subfamilies in total. The *LBD* gene family in *P. bournei* (Hemsl.) Yang species had two subfamilies, including 32 genes in Class I and 6 genes in Class II. It mainly consists of a Lateral Organ Boundary (LOB) conservative domain, and the protein structure is mostly “Y”-shaped. The gene expression pattern of the *LBD* gene family showed that the *LBD* genes were mainly expressed in lateral organs of plants, such as flowers and fruits. The response of LBD transcription factors to red and blue light was summarized, and several models of optogenetic expression regulation were proposed. The effect of regulatory mechanisms on plant rooting was also predicted. Moreover, quantitative real-time PCR (qRT-PCR) revealed that most *PbLBD*s were differentially expressed under cold, heat, drought, and salt stresses, indicating that *PbLBD*s might play different functions depending on the type of abiotic stress. This study provides the foundation for further research on the function of LBD in this tree species in the future.

## 1. Introduction

*Phoebe bournei* (Hemsl.) Yang is a nationally conserved tree species in China and is known for its stress resistance. It possesses high economic value and has a positive effect on the ecological environment by enhancing soil fertility and maintaining water, making it a valuable resource for the country [1,2]. This species belongs to the *Phoebe* genus and exhibits an ideal wood structure for both industrial and domestic applications, making it economically valuable [2,3]. Inducing rooting during cultivation remains a challenging task. Exploring the genes in *P. bournei* that promote root development and respond to external abiotic factors may help solve this problem.

Transcription factors (TF), also known as trans-acting factors, are special proteins with unique structures that enable them to bind to specific DNA sequences in gene promoters. Through this binding, they regulate the expression of target genes [4], which in turn affects plant growth. These transcription factors play a crucial role in controlling the differentiation and proliferation of cells and have been identified in various plant species [5]. Due to the complex and variable morphology of plants, influenced by both their external environment and internal genetics, different transcription factors are involved in regulating the development of different tissues, such as leaves [6,7]. 

The Lateral Organ Boundary Domain (LBD) family is a group of transcription factors that play an important role in lateral organ formation and secondary growth of plants, including promoting root growth [8]. They originated from early embryonic aquatic plants, and during plant evolution, two gene amplification events primarily occurred during the transition from aquatic to terrestrial plants and from lower to higher plants [8]. LBD members are found in most angiosperms, with over 20 family members in plants, and they have well-established regulatory effects [5,6,9]. Most LBD members possess a unique conserved LOB domain of approximately 100 amino acid residues, consisting of a C-terminal motif (C-block), Gly-Ala-Ser motif (GAS-block), and a leucine-zipper-like motif (leucine-zipper-like block) [10,11]. The C-block contains conserved cysteine residues in the CX2CX6CX3C motif that are required for DNA binding activity, spanning 22 amino acid residues. The GAS-block consists of 49 amino acid residues with a conserved proline residue that affects DNA binding activity. The leucine-zipper-like block, composed of 19 amino acid residues, is separated by four conserved leucine residues and non-conserved X residues, forming the LX6LX3LX6L structural pattern, which is associated with dimerization. The variable C-terminus is connected to the conserved leucine-zipper-like block and regulates the expression of downstream genes, thereby constituting the genetic structure of *LBD* [12]. These features highlight the functional and structural diversity of *LBD* members and their important role in regulating plant development. In the study of *Panicum virtum* L., 69 members of the LBD family were identified from the entire genome sequence. Based on their homology with the Arabidopsis *LBD* gene, 62 cis-acting element *PvLBD* genes were selected that can respond to light, thereby controlling plant growth, regulating circadian rhythm, and responding to stress [13]. AtLOB is a specific LBD member that binds to the DNA sequence “GCGGCG” through the Lateral Organ Boundary (LOB) domain, which possesses a nuclear localization function.

Based on the differences in their conserved domains, the LBD family is divided into two subfamilies: Class I contains three motifs, namely the C-domain, GAS-block, and leucine-zipper-like motif, while Class II has an incomplete leucine-like zipper motif [11,13]. The motifs of Class II are relatively conserved, with high similarity among members, but they cannot form a coiled coil structure. LBD proteins have been identified in many plant species, including *A. thaliana*, rice, and *R. longa* [10,11]. 

To better understand how plants respond to changes in their environment, various physiological and biochemical indicators are used to reflect their life activities. These indicators include chlorophyll content, leaf area index, and enzyme activities, all of which are influenced by gene expression [14,15]. Therefore, studying the expression of related gene families can provide insight into how plants respond to different conditions. Additionally, physiological and biochemical indicators related to photosynthetic capacity can more accurately reflect the levels of synthetic substances and energy in plants [13]. Chlorophyll, in particular, is crucial for plant photosynthesis and plays a significant role in the metabolic processes of the entire plant [1,14].

Previous research has also demonstrated the significant role of *LBD* genes in enhancing stress resistance in plants. For example, the *RrLBD25* gene from *Rosa rugosa* has been found to exhibit a negative response to salt stress signals, possibly through cross-talk with auxin signaling [15]. In *Solanum tuberosum*, genes *StLBD1-4* and *StLBD9-1* were significantly upregulated under salt stress, indicating their involvement in salt tolerance response [16,17]. Moreover, *StLBD1-2* and *StLBD4-1* showed high expression levels under heat and drought stresses, suggesting that *LBD* genes may also aid plants in combatting the threats posed by these stressors. Furthermore, a study on bananas demonstrated that *LBD* genes jointly respond to cold stress and *MeJA* (methyl jasmonate) treatment, thereby alleviating chilling injuries. In this process, physical interactions between *MaLBD5* and *MaJAZ1* may attenuate the transactivation of endogenous jasmonic acid (JA) [18] that intensifies cold stress, as illustrated in Figure 1. These suggest that *LBD* genes have broad implications for stress resistance in plants. The ability of these genes to enhance plant resilience against multiple stressors highlights their potential as targets for crop improvement and breeding strategies, thereby contributing to food security and sustainable agriculture. However, the *LBD* gene family in *P. bournei* has not been reported to date.

Current research on the LBD family and its impact on plant physiology and biochemistry has been the focus of numerous studies. Research on the effects of the LBD family under different light conditions is still lacking. The objective of this study is to identify the *LBD* genes in *P. bournei* and investigate their response to abiotic stress and various light qualities. Additionally, this study aims to examine how the LBD family affects the rooting process of *P. bournei*. By investigating these aspects, we hope to provide new insights into the role of the LBD family in *P. bournei* and contribute to the development of more effective cultivation processes for this plant species.

## 2. Results

### 2.1. Identification of LBD Family Members and Their Physicochemical Properties

Homology comparison with *Arabidopsis thaliana LBD* genes led to the identification of 38 putative *LBD* gene sequences in *P. bournei*. An online tool, ExPASy, was used for sequence analysis, revealing the length of the PbLBD proteins ranged from 120 to 412 amino acid (AA) residues. The molecular weights ranged from 13.44 to 46.40 kDa and the theoretical isoelectric points (PI) varied from 4.94 to 9.49. All PbLBD protein sequences, except for PbLBD9, had a negative Grand Average of Hydropathicity (GRAVY), indicating that PbLBD proteins were predominantly hydrophilic. Subcellular localization analysis indicated that these proteins were primarily located in the stroma of mitochondria and chloroplasts. This suggested that this transcription factor family could play a vital role in plant growth, development, and metabolism (Table 1).

### 2.2. Phylogenetic Analysis of the LBD Family Members in P. bournei

To investigate the evolutionary relationship of *PbLBD* genes, a comparison was made between the *LBD* genes from *P. bournei* and *A. thaliana*. It is worth noting that the expression of the *LBD* gene family in plants is widespread and relatively conserved [13,19]. To confirm the homology between the *LBD* genes of these two species, a phylogenetic tree was constructed using the corresponding protein sequences (Figure 2). The *LBD* genes were divided into two subclasses: Class I, which consisted of 32 different *LBD* genes, and Class II, which consisted of 6 different *LBD* genes.

Cluster analysis of the phylogenetic tree was carried out to analyze the distribution of *PbLBD* and *AtLBD* genes. Within each subclass of Class I, there were seven *PbLBD* genes and thirteen *AtLBD* genes in subclass Ia, nine *PbLBD* genes and nine *AtLBD* genes in subclass Ib, nine *PbLBD* genes and five *AtLBD* genes in subclass Ic, and seven *PbLBD* genes and ten *AtLBD* genes in subclass Id. The results of the analysis indicated that each subclass encompassed related genes from both *P. bournei* and *A. thaliana*, suggesting a shared ancestral origin for these genes. Class II had two branches: IIa and IIb. IIa consisted of five gene family members from *P. bournei* and six members from *A. thaliana,* while IIb had one gene family member from *P. bournei* and none from *A. thaliana*. Notably, *PbLBD37,* a single *LBD* gene in Class II, had no counterpart in *A. thaliana.* The counterparts of PbLBD proteins in *Arabidopsis* can be found in the Appendix A. The difference might be attributed to the long-term evolution of these two species. Furthermore, it was also speculated that functional differentiation might have occurred during the evolution process due to the loss of certain *LBD* genes in *Arabidopsis* species. Through a phylogenetic comparison of *LBD* genes in various plants, insights could be gained into the evolutionary expansion and conservation of these genes [12,20].

### 2.3. Analysis of Conserved Motifs and Protein Structure of PbLBD

A total of 10 conserved motifs were identified through an analysis of conserved domains and structures in *P. bournei* LBD proteins (Figure 2). Two conserved domains, LOB and AP2, were present in all LBD members in *P. bournei*. Furthermore, it was confirmed that LOB conserved domains primarily consisted of Motif 1, Motif 2, Motif 3, and Motif 4, which were complete in all 31 Class I members (Figure 3). 

In the case of the six members of *P. bournei* Class II, it was observed that most conserved domains of the LOB members were only partially present. This could be a result of genetic differentiation or other events that led to the loss of these conserved domains. However, it is interesting to note that all Class II members contained Motif 5, suggesting that Motif 5 might play a crucial role in determining the protein specificity of Class II. As shown in Figure 3, Motif 9 was present in all Class II LBD members except for PbLBD38, and was absent in Class I. PbLBD38 also did not contain Motif 2. The absence of conserved motifs 9 and 2 in PbLBD38 raised the possibility that PbLBD38 might not be functional. Further extensive research is required to validate this speculation.

### 2.4. LBD Protein Sequence Comparison and Their Tertiary Structure

A multiple sequence comparison of 38 PbLBD proteins was performed using the software DANMAN (version 9) (Figure 4), resulting in the identification of two conserved protein domain sequences. All members from the two subclasses of PbLBD shared a conserved domain CX2CX6CX3C. However, only 10 sequences, namely PbLBD2, PbLBD11, PbLBD12, PbLBD15, PbLBD17, PbLBD19, PbLBD21, PbLBD22, PbLBD24, and PbLBD30, contained a second conserved LX6LX3LX6L leucine-zipper-like domain. The remaining members only possessed a partial domain. The overall composition of the leucine-zipper-like domain was V81%I61%X6V35%X6L50%X6L60%. The online software SWISS-MODEL (https://swissmodel.expasy.org/interactive) predicted the optimal structure of the LBD family in *P. bournei* based on the GMQE value closest to one, and it also predicted the structures of the members from the two subclasses. The modeling analysis revealed that the spatial 3D structure of each related LBD exhibited a symmetric “Y”-like shape (Figure 5). It was speculated that due to the homology of protein sequences and conserved domain structure at both the C- and N-terminus, these two subfamilies shared functional similarities.

To gain a better understanding of the biological function of LBDs in *P. bournei* and their interactions, we predicted the interaction pattern using the online tool String. The network nodes represent folding or post-translational modifications of protein splicing isomers. Each node represents all proteins produced at a single protein site, while edges represent protein–protein associations. These associations suggest specific and meaningful protein interactions that contribute to a shared function. As shown in Figure 6, LBD proteins interacted with the KNOX (knotted-like homeobox) family and LOB domain-containing proteins. The KNOX family plays a vital role in plant development and physiology, consisting of multiple transcriptional regulatory factors [21]. For instance, the *ASYMMETRIC LEAVES 1 (ASL1)* gene encodes a LOB domain protein. Overexpression of *ASL1* results in the repression of the *KNOX* gene, and leads to phenotypes such as hyponastic leaf, downward pointing flower, and reduced apical dominance [22]. The interaction network among LBD proteins and other protein families, as shown in Figure 6, reveals that the physiological functions of proteins encoded by *LBD16* and *LBD18* were notably varied and complex. LBD18 is involved in the positive regulation of Tracheary Element (TE) differentiation, and functions in the initiation and emergence of lateral roots. LBD16, on the other hand, is downstream of the AUXIN RESPONSE FACTOR7 and 9 (ARF7 and ARF9) in the network [21]. LBD16 is a transcriptional activator that directly regulates *EXPANSIN A14 (EXPA14)*, a gene encoding a cell wall-loosening factor, and promotes lateral root emergence. 

### 2.5. The Variance of LBD Gene Expression Level in Different Tissues in P. bournei

An analysis of gene expression patterns revealed a significant correlation between the expression levels of *LBD* gene expression and their respective functions. The primary focus of this study was to examine the gene expression levels in different tissues, such as stamen, pistil, tepal, petal, fruit, leaf, xylem, and radicle, as shown in Figure 7. By comparing the gene expression across these tissues, it was found that there were variable levels of expression, as illustrated by a heat map with a color spectrum ranging from blue (indicating low expression) to red (indicating high expression). It was observed that the majority of gene family members exhibited higher expression levels in radicles, flowers, and tepals. This suggested that this particular gene family could have a significant impact on the development and formation of flowers and other related organs.

It was also observed that some *LBD* family genes exhibited high expression levels in leaves, implying that these genes could play a key role in regulating photosynthesis in plants. It is worth noting that the blue color on the heat map does not necessarily indicate the absence of gene expression in the tissue, but rather a low level of expression. This low expression might be due to environmental stress or developmental stages. Generally, genes with different expression levels regulate the growth and development of various plant tissues. This study revealed that genes *PbLBD1/7/21* had the highest expression in fruit, *PbLBD16/17/28* had the highest expression in leaf, and gene expression was generally higher in radicle but lower in xylem (Figure 7).

### 2.6. The Induced Expression of PbLBD Genes Associated with Stress Treatment 

Based on the gene family characteristics discussed in the previous section, we selected five genes (*PbLBD16/17/20/26/28*) that exhibited higher expression levels in lateral organs such as leaves and flowers. We then verified the differences in expression levels in response to temperature, salt, and drought stress were then verified through qRT-PCR analysis (Figure 8). The expression patterns of *PbLBD*s at 10 °C indicate that *PbLBD16/17/20/26/28* were all induced by low temperature, and overall exhibited an “up-down-up” trend. However, the expression levels at 40 °C suggested a generally negative response to heat stress across the gene family, except for *PbLBD28*, which showed higher expression after 24 h at 40 °C. This suggested that PbLBD28 might play a critical role in coping with heat stress. 

Under salt stress, *PbLBD16/17/20/26* resisted stress effectively during the 4 h treatment but showed almost no expression thereafter. *PbLBD28* exhibited a decreasing expression trend, indicating a generally low salt tolerance within this gene family. In response to drought stress (10%PEG), *PbLBD17* displayed the most robust performance, while the remaining genes generally showed decreased expression levels. This revealed the importance of *PbLBD17* in drought stress response. Our qRT-PCR analysis revealed that within the *PbLBD* gene family in *P. bournei*, genes with more pronounced phenotypes were primarily regulated by cold stress, indicating their active role in coping with low temperatures.

## 3. Discussion

### 3.1. The Role of the Transcription Factor for Lateral Organ Boundary Domains in Phoebe bournei and Its Significance for Bioinformatics Research

The *LBD* gene family, which encodes a conserved LOB domain, is a plant-specific transcription factor family that plays a vital role in the growth, development, and formation of lateral organs such as blossoms and leaves. The *LBD* gene family is also involved in regulating stress response [9,23]. Although previous studies have identified *LBD* genes in various species including *Arabidopsis* [24], grape [25], and passion fruit [20], no research has been conducted on the *LBD* gene family in *P. bournei* until now.

The C-terminus is a variable section that regulates the expression of downstream genes, and it is connected to the conserved leucine-zipper-like block, forming the genetic framework of LBD [12]. These characteristics highlight the structural and functional diversity present in LBD members and underscore the importance of their function in controlling plant growth. In this study, we compared the *AtLBD* genes with the *P. bournei* genome and identified a total of 38 related *LBD* genes in this species. Phylogenetic analysis of *LBD* gene families from both *A. thaliana* and *P. bournei* revealed their evolutionary relationships. This analysis allowed us to classify the *PbLBD* genes into two classes, both of which exhibited specific LOB domains. However, it is worth noting that most members of Class I contained complete LOB domains, whereas Class II members had domains with deletions, suggesting domain loss may have occurred during the evolutionary process [23]. All 38 identified *LBD* protein sequences in *P. bournei* exhibited the structural pattern CX2CX6CX3C, but only a subset of them had the LX6LX3LX6L motif [26]. Among them, 10 protein sequences were specifically identified in Class II, with Class I sequences exhibiting the LX6LX3LX6L motif, suggesting a close relationship between protein dimerization and their functional structure.

According to our analysis (Figure 3), all identified *LBD* gene family members in *P. bournei* possessed a LOB conserved domain, indicating their functional similarity. Meanwhile, most of them contained four conserved motifs that aligned with the LOB domain, suggesting that these four conserved motifs formed the core composition of the LOB domain [27]. Previous research on the LBD gene family has shown that it may be separated into two classes: Class I, which has complete CX2CX6CX3C and LX6LX3LX6L motifs, and Class II, which contains a complete CX2CX6CX3C motif and a residual LX6LX3LX6L motif. In this study, *LBD* gene family members were also separated into two classes. Class I had 32 genes, which were further divided unevenly into four subclasses according to the evolutionary tree’s branch clustering relationship, while Class II had six genes which were divided into two subclasses. All Class II members contained Motif 5. Motif 9 was present in all Class II LBD members except for PbLBD38 and was absent in Class I. These findings suggest that the evolution of the *LBD* gene family in *P. bournei* is primarily conservative. Analysis of the divergence relationship of the *LBD* family in the model species Arabidopsis and the phylogenetic tree of *PbLBD* genes identified five distinctive subgroups. Moreover, genes with different conserved motifs generally clustered in different branches, suggesting that the *LBD* gene family members within the same subfamily branch may have similar regulatory functions. Comparable findings are seen in Liang et al. (2022), which suggests that the results share comparable functional properties as indicated by the same domain and motif [20]. Additionally, we created 38 alternative protein modeling scenarios for *P. bournei,* which can depict functional similarities.

### 3.2. The Effects of Environmental Factors on LBD Genes in P. bournei

#### 3.2.1. The Consequences and Mechanism of How the Transcription Factors Respond to Varying Light Quality

Plants possess a diverse array of photoreceptors that have the ability to perceive and respond to various light characteristics. Among these photoreceptors, phytochromes have been extensively investigated in the model plant *A. thaliana.* Phytochrome is involved in numerous physiological processes, including seed germination, seedling development, photosynthesis, flowering, and shade avoidance [28]. Upon exposure to red light, phytochrome proteins translocate from the cytoplasm to the nucleus, where they actively interact with transcription factors, including Phytochrome Interacting Factor (PIF) [29]. These interactions provide valuable insights into the impact of different light qualities on transcription factors during the initial stages of *P. bournei* cutting rooting.

While the previous literature has not elucidated the mechanism by which light regulates *P. bournei* gene expression in vivo, a model can be inferred by comparing gene expression patterns in other species in response to light qualities. To investigate this, we conducted transcriptome analysis to examine the effects of red, blue, and white (control) light qualities on *P. bournei* growth. Analyzing the resulting heat map of gene expression can provide a reference for constructing gene expression models in plants affected by different light qualities. Although there is no research on how light quality impacts the organ growth mechanism of *P. bournei*, we propose a photomorphogenesis model based on our results (Figure 9).

Given the structural similarity of the transcription factors involved in this study to those in *Arabidopsis*, it is plausible that similar physiological and biochemical responses occur in *P. bournei*. Under our proposed model, P_fr_A and P_fr_B can directly interact with G proteins on the cell membrane [30], thereby activating nuclear transcription factors (*LBD* gene family) via cGMP or Ca^2+^-CaM, and regulating gene expression. Similarly, cryptochrome-mediated blue light signaling might affect chloroplast gene transcription under blue light conditions [33]. As indicated by previous studies on *Arabidopsis* [31,32,34], both cryptochromes (CRY1, CRY2) and calcium ions in *P. bournei* can regulate chloroplast gene expression in response to blue light, thereby affecting various physiological and life activities of the plant.

#### 3.2.2. The Response of Genes under Cold, Heat, Salt, and Drought Stress

Previous studies have confirmed that multiple *LBD* genes can be simultaneously induced and expressed in plants under extreme stress, thereby participating in stress response regulation. For example, overexpression of tomato (*Lycopersicon esculentum*) *SILBD40* under drought stress causes severe wilt in the plants, while gene knockout leads to slight wilting [26]. In a study by Huang et al. [35], *Pyscomitrella* treated with mannitol showed increased expression of most *PpLBDs* genes, implying their potential role in enhancing plant drought tolerance. Under drought stress, *AtLBD15* can directly bind to the promoter of the *ABII4* signaling pathway factor to promote stomatal closure, thus reducing water loss and improving plant drought resistance [36]. Liu et al. conducted qRT-PCR analysis on potato (*Solamum tuberosum*) after drought stress and found that the expression of *StLBD1-5* was decreased while that of *STlBD2-5* and *StLBD3-5* was up-regulated, indicating their involvement in potato drought response [16]. The induced expression of *LBD* genes can activate the expression of stress-related genes and regulate plant adaptive response in the stress regulatory network. Such transcription factor response is generally rapid and instantaneous. Combined with phylogenetic analysis, transcriptome data, and qRT-PCR analysis, this study verifies that five *PbLBDs* genes can be rapidly induced and activated in response to abiotic stress such as low phosphorus, drought, and salt stress, thus functioning in abiotic stress.

In this study, we analyzed the transcriptome data of *P. bournei* seedlings subjected to abiotic stress conditions such as cold, heat, drought, and salt. We found that the expression levels of five *LBD* genes significantly varied depending on stress duration. These results indicate that *PbPBD* genes are induced by stress and participate in the stress response regulatory network. Although five genes showed significantly lower expression levels under heat stress (40 °C), most of them displayed increased expression levels under low temperatures, drought, and salt stress. In particular, under salt stress (10% NaCl), genes *PbLBD16/17/20/26* were rapidly induced, showing an increased expression pattern at 4 h, suggesting their potential role in stress-related gene expression under salt stress. Based on these findings, we predicted that these *PbPBD* genes may play a positive regulatory role in *P. bournei* abiotic stress.

## 4. Materials and Methods

### 4.1. Materials

#### 4.1.1. Plant Materials and Treatment

Since the LBD family is involved in inducing lateral root formation in plants, *P. bournei* epidermis genes were selected for transcriptome sequencing. The genome sequence for *P. bournei* was provided by Dr. Shipin Chen. We utilized protein sequences encoded by *LBD* genes from the *Arabidopsis* database (https://www.arabidopsis.org, accessed on 4 May 2022) for genetic screening and identification. *P. bournei* seedlings were exposed to different light treatments. White light (RGB intensity was R: 150, G: 150: B: 150) experimental group was established one week in advance in the light quality chamber, followed by a 48 h observation period with sampling at six-hour intervals, starting from 12:00 Beijing time. During this period, the light intensity of the chamber was set to R: 000, G: 000, B: 150, with measured illuminance of approximately 80.5 lux. The specimens were disinfected and stored in liquid nitrogen throughout the sampling process, and stored at −80 °C for subsequent experiments. A spectrophotometer, electronic balance, 95% ethanol solution, quartz sand, calcium carbonate, and a brown volumetric flask were prepared for chlorophyll determination.

#### 4.1.2. Test Setup, Parameter Settings, and Processing Instructions

The experimental apparatus was set up as an intelligent cutting box purchased from a Fuzhou-based company, featured with three independent layers with shading films and an external light-blocking door. The intelligent system control panel is ensured consistency in other variables such as light intensity, temperature, and humidity. The parameters of the light incubator were set as follows: the temperature set to 30 ± 2 °C, humidity at 90%, light intensity at 50 μmol·m^−2^·s^−1^ through the top of the acrylic plate cutting, and photoperiod of 16h light/8h dark. The LED light qualities included white light (W) as the control, red light (R, wavelength 660 nm), and blue light (B, wavelength 460 nm) as treatments, with each incubator layer accommodating different light conditions.

Three stages of rooting in *P. bournei* cuttings were selected and observed, namely T0, T1, and T2 stages. At each stage, the vascular formation of terminal buds, leaves, and stem segments within 2 cm from the base of cuttings under different light treatments was examined. Applying the whole epidermis outside of the vascular cambium, seven samples were collected from each part and snap-frozen in liquid nitrogen for storage at −80 °C until further use. The samples included Pb_A_T0, Pb_AB_T1, Pb_AB_T2, Pb_AR_T1, Pb_AR_T2, Pb_AW_T1, and Pb_AW_T2 of apical buds, where ‘Pb’ represented *P. bournei*, ‘A’ represented apical bud, ‘B’, ‘R’, and ‘W’ indicated blue light, red light, and white light treatments, respectively, and T0, T1, and T2 represented three periods (L stood for leaves and P for phloem). 

### 4.2. Methods

#### 4.2.1. Identification of the LBD Gene Family in *P. bournei*

To identify LBD domain-containing genes in the *P. bournei* genome, LBD protein sequences from the *Arabidopsis* gene family were downloaded from the TAIR database for comparison (https://www.arabidopsis.org/, accessed on 4 May 2022). The e-value threshold was set to 1 × 10^−5^ otherwise the default parameter was chosen for the alignment of Arabidopsis LBD protein sequences to *P. bournei* protein sequences using search tools (https://github.com/CJ-Chen/TBtools/releases, Tbtools version 1.108, accessed on 12 April 2022). The Pfam file, downloaded from the Pfam protein family database (https://pfam.xfam.org/, accessed on 4 May 2022) [37] was combined with the screening sequence to construct a hidden Markov model and screened candidate sequences containing PF03195 protein sequence, a Pfam domain. The LBD conserved domain’s presence in each of the selected candidate genes was subsequently confirmed using the NCBI-CDD search tool (https://www.ncbi.nlm.nih.gov/Structure/bwrpsb/bwrpsb.cgi, accessed on 4 May 2022), the SMART network database (http://smart.embl-heidelberg.de/, accessed on 30 April 2023), and the Interpro online website (http://www.ebi.ac.uk/interpro/search/sequence/, accessed on 30 April 2023). TBtools version 1.108 was applied in this process.

#### 4.2.2. Determination of Physicochemical Properties of PbLBD Genes and Protein Prediction

The physicochemical properties of the deduced PbLBD proteins from the selected genes were identified using the online software Expasy (https://web.expasy.org/cgi-bin, accessed on 4 May 2022). This allowed for the determination of essential parameters such as isoelectric point. SWISS-model homology modeling was used to predict the protein 3D structure of the LBD gene based on GMQE value. The subcellular localization of LBD proteins was predicted using the online tool WoLF PSORT (https://wolfpsort.hgc.jp/, accessed on 16 May 2022), while the secondary structure of LBD proteins was predicted using PRABI (https://npsa-prabi.ibcp.fr/cgi-bin/, accessed on 30 April 2023).

#### 4.2.3. Phylogenetic Analysis of the LBD Gene Family in *P. bournei*

MEGA7.0.26, MUSCLE, and IQ-tree were used to construct an evolutionary tree from *P. bournei* and *Arabidopsis Thaliana* based on the maximum likelihood method and local comparison. The MUSCLE software (version 3.8.31) was used for protein sequence alignment, and the IQ-TREE multicore software (version 1.6.10) was used to set the local comparison based on the maximum likelihood method, and the parameter was 1000 repetitions to build the evolutionary tree, and then the visualization of the phylogenetic tree was carried out through the MEGA software (version 7.0.26 (7170509-x86_64)).

#### 4.2.4. Analysis of the Conserved Motifs, Gene Structures, and Characteristic Domains of the LBD Gene Family in *P. bournei*

The online tool MEME (http://meme-suite.org, accessed on 4 May 2022) was used to predict the characteristic domains and conserved motifs of the LBD family in *P. bournei,* using the protein sequences previously aligned via the NCBI website. The repetition setting was set to Zero or One Occurrence per Sequence (ZOOS), and the maximum motif repetitions were set to 10 and 3, to identify relevant motifs with special domains. Subsequently, the protein domain was determined by using TBtools software (version 1.108, accessed on 3 May 2022). Finally, the related protein sequences encoded by *PbLBD* genes were aligned by using DNAMAN software (version 9), enabling the analysis of conserved domains of the lateral organ boundary domain genes.

#### 4.2.5. Expression Patterns of *PbLBD* Gene

Transcriptome data was provided by Dr. Shipin Chen’s team. From this transcriptome data, the expression of the identified *LBD* gene family in *P. bournei* was extracted. Heat maps were generated by TBtools software to illustrate the expression patterns.

A mature P. bournei tree growing at the Fujian Agriculture and Forestry University in the Chinese province of Fujian served as the source of all the plant samples utilized in this investigation. For Illumina and de novo sequencing and assembly, whole genomic DNA was extracted using a modified cetyltrimethylammonium bromide (CTAB) technique. Errors in the raw data are corrected by Canu52. Gene prediction and annotation were applied using Exonerate v2.2.0 (https://www.ebi.ac.uk/Tools/psa/genewise/, accessed on 1 September 2020) [38]. From Phytozome 12 (https://phytozome.jgi.doe.gov/pz/portal.html, accessed on 1 September 2020), the sequences of these known genomes were retrieved. De novo gene prediction was performed using two ab initio prediction software tools, Augustus [39] (http://bioinf.unigreifswald.de/augustus/, accessed on 1 September 2020) and SNAP62 (http://homepage.mac.com/iankorf, accessed on 1 September 2020).

After that, a nonredundant gene model was created using Maker [40] (http://weatherby.genetics.utah.edu/MAKER/wiki/index.php/MAKER_Tutorial_for_WGS_Assembly_and_Annotation_Winter_School_2018, accessed on 1 September 2020) by combining the homology-based and ab initio gene structures. The following genes were removed from the Maker annotated results: (1) proteins with lengths less than 50 amino acids and homologous protein support for exon regions below 50%; and (2) those with CDS of the coding region and TE overlap lengths more than 80%. From TAIR (www.arabidopsis.org/index.jsp), the candidate sequences for the MADS-box and SAUR genes of A. thaliana were retrieved. Each protein from the gene families of *P. bournei* and *C. kanehirae* was individually searched in the genomes using the HMMER 3.2.1 (with default parameters) [41] and BLASTP (E-value of 1 × 10^−5^) [42] techniques by Kent et al. Horticulture Research (2020) 7:146 Page 11 of 13 [43].

#### 4.2.6. Protein Interaction Network Analysis of *PbLBD* Proteins

The interaction relationships of 20 relevant proteins were predicted with the online tool String (https://cn.string-db.org/, accessed on 14 July 2022), and a protein interacting network was subsequently constructed. Based on known homologous proteins encoded in *Arabidopsis* as a reference, the STRING database (https://stringdb.org, accessed on 14 July 2022) was applied to predict and analyze the interaction network of LBD domain proteins. The following parameters were set for the STRING database: network type was set to full STRING network, the network edges were evidence-based, and a medium confidence parameter (0.4) was used to represent the minimum interaction score. To ensure data accuracy, the maximum number of interactions was capped at 10.

#### 4.2.7. Plant Material and Stress Treatment with RNA Extraction and qRT–PCR

The materials were derived from one-year-old seedlings provided by the Fujian Academy of Forestry. Seedlings were cultured in an artificial climate chamber and subjected to different treatments. *P. bournei* seedlings of similar growth potential were selected and divided into a control group and a stress treatment group. Each treatment group consisted of 30 individuals while the control group contained 3 individuals. After treatments, leaf samples were collected and stored in liquid nitrogen at −80 °C for subsequent RNA extraction. For the drought stress simulation, the control group seedlings were soaked in distilled water, whereas the treatment group was soaked in a nutrient solution containing 10% PEG. Another group for salt treatment was soaked in a nutrient solution containing 10% NaCl. For temperature stress treatment, the control group was maintained at room temperature, while corresponding treatment groups were incubated at 40 °C, or 10 °C. 

All samples were cultured in an artificial climate chamber with a temperature of 25 °C and a humidity of 75%. The treatment group was sampled at 4, 6, 8, 12, and 24 h, respectively, and the control group was sampled at 2 h. 

RNA was extracted with HiPure Plant RNA Mini Kit (Med-based Bio). Subsequently, cDNA was synthesized by a Prime Script RT reagent Kit (Perfect Real Time) from TaKaRa. Based on cis-acting elements present in *P. bournei* LBD family members, *PbLBD16/17/20/26/28,* which contain the most elements related to stress response, such as low temperature, high temperature, NaCl, and PEG, were selected for qRT-PCR analysis. Specific primers (Table 2) were designed in the non-conserved region of the target gene using Primer3.0 software and synthesized by Fuzhou Qingbaiwang Biotechnology Company (Fuzhou, China).

Real-time PCR analysis was performed with 1 μL cDNA template, 10 μL SYBR Premix Ex Taq^TM^ II, 2 μL specific primers, and 7 μL ddH_2_O. The reaction conditions were as follows: 95 °C for 30 s; 95 °C for 5 s; 60 °C for 30 s; 95 °C for 5 s; 60 °C for 60 s; and 50 °C for 30 s, for a total of 40 cycles. The internal reference gene was *PbEF1α* (GenBank No. KX682032) [44]. The expression level of the target gene was calculated using the 2^−ΔΔCt^ method, and the quantitative data were analyzed by *t*-test in SPSS26. Statistical significance was determined when *p* < 0.05. GraphPad Prism8.0 was applied for data visualization.

## 5. Conclusions

A total of 38 *LBD* genes were identified in *P. bournei* through diversification analysis in this study. Their physicochemical properties, conserved gene structures, phylogenetic relationships, and expression patterns were assessed. The *PbLBDs* could be divided into five distinctive branches based on the comparison of evolutionary relationships, akin to those found in *Arabidopsis*. This indicated that the gene family in *P. bournei* may perform similar functions to those in *Arabidopsis*. This conclusion aligns with the evidence of conserved development found in the previously verified conserved domains and motifs. Multiple sequence alignment with significant similarities further infers that the LBD family plays a role in various plants or has high consistency in those plants. Additionally, based on FPKM values and transcriptome data, we analyzed the expression of this gene family in different tissues and found that 35 different members of the LBD family had differential expression in various tissues. The expression analysis of qRT-PCR showed that the LBD gene family was regulated by cold stress and had a positive response to low-temperature stress. In addition, we found that *PbLBD* members showed negative expression patterns under both salt stress and drought stress, compared with the control group. This study systematically analyzed the expression patterns of *LBD* genes in different tissues and their response to abiotic stress, which was helpful to further reveal the mechanism of *LBD* genes in the stress response signaling pathway of *P. bournei*, and provided new insights and information for future research on species selection and stress resistance regulation of *P. bournei*.

## Figures and Tables

**Figure 1 ijms-24-12581-f001:**
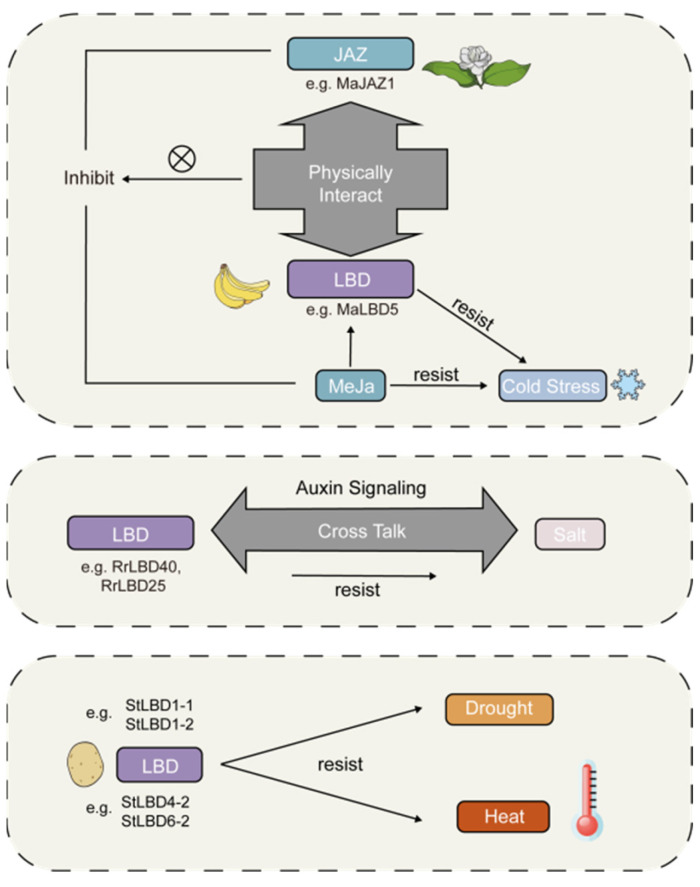
Abiotic stress response mechanisms of the *LBD* gene in different species. The diagrams provide an overview of how the *LBD* gene helps different plant species reduce injury caused by abiotic stress.

**Figure 2 ijms-24-12581-f002:**
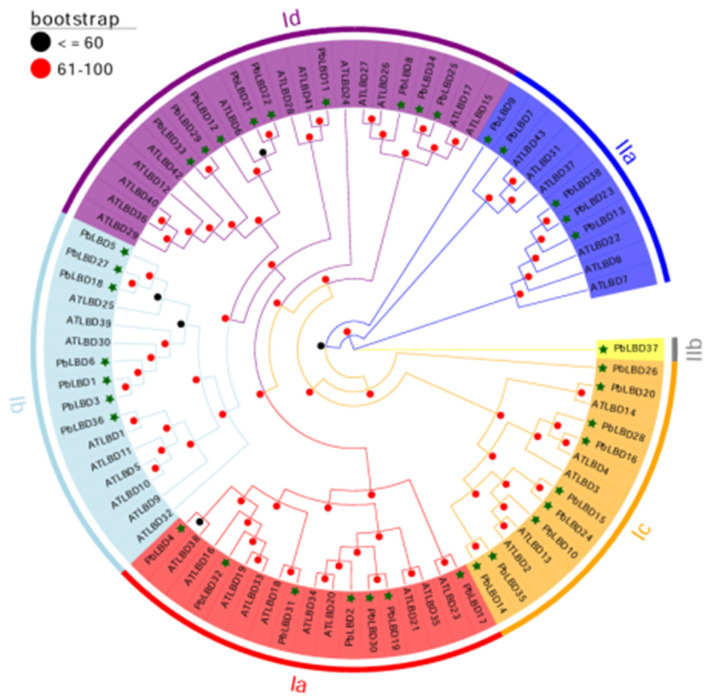
The phylogenetic tree of the boundary domain genes of the lateral organs in *P. bournei* and *A. thaliana*. The asterisk represent the members of *P. bournei*. The numbers on the branch represent the reliability of the node with 1000 bootstrap. Branches of different categories are in different colors. Each LBD subclass is indicated by an arc. As indicated in the phylogenetic tree clustering structure, Class I was further classified into subclasses Ia, Ib, Ic, and Id; Class II includes subclasses IIa and IIb.

**Figure 3 ijms-24-12581-f003:**
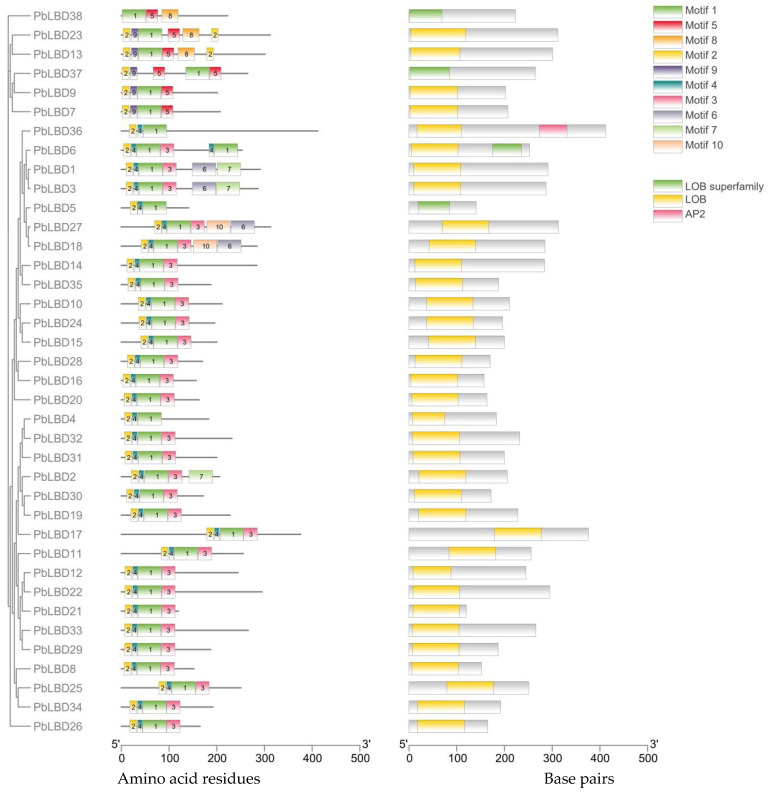
The phylogenetic tree and conserved motifs, domains of the *P. bournei* LBD family and LOB superfamily. On the left side is the neighbor-joining tree comprising 38 motifs. Conserved motifs were represented by viaboxes, and different colors represent different motifs. On the right side, yellow boxes represent LOB domains and green ones represent LOB superfamily.

**Figure 4 ijms-24-12581-f004:**
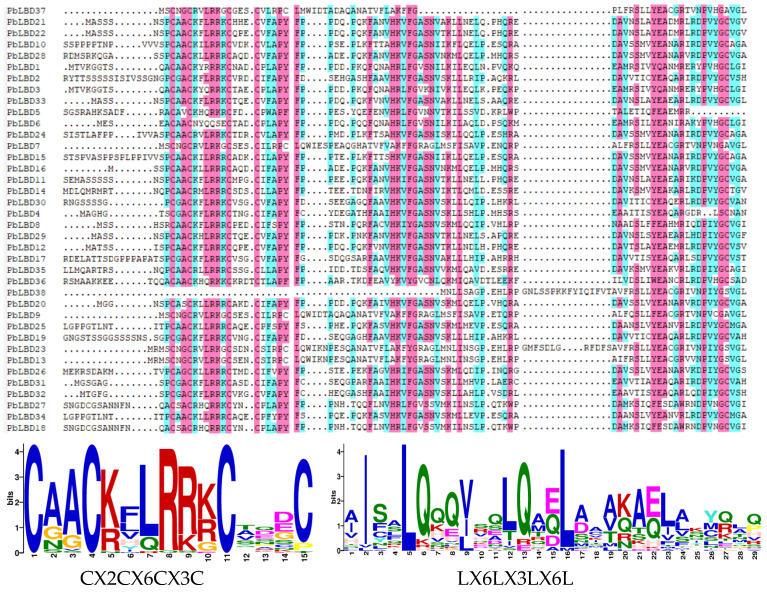
Sequence alignment of conserved protein domains within the boundary domain family in the lateral organs of *P. bournei*. The CX2CX6CX3C zinc finger-like domain was present in the 38 proposed PbLBD protein sequences, while a complete LX6LX3LX6L domain was only presented in some sequences. The other sequences contained only a partial LX6LX3LX6L domain. The conserved motif logos were generated with different colors. The blue and purple highlights were applied to identify the unique properties of the same conserved sequences.

**Figure 5 ijms-24-12581-f005:**
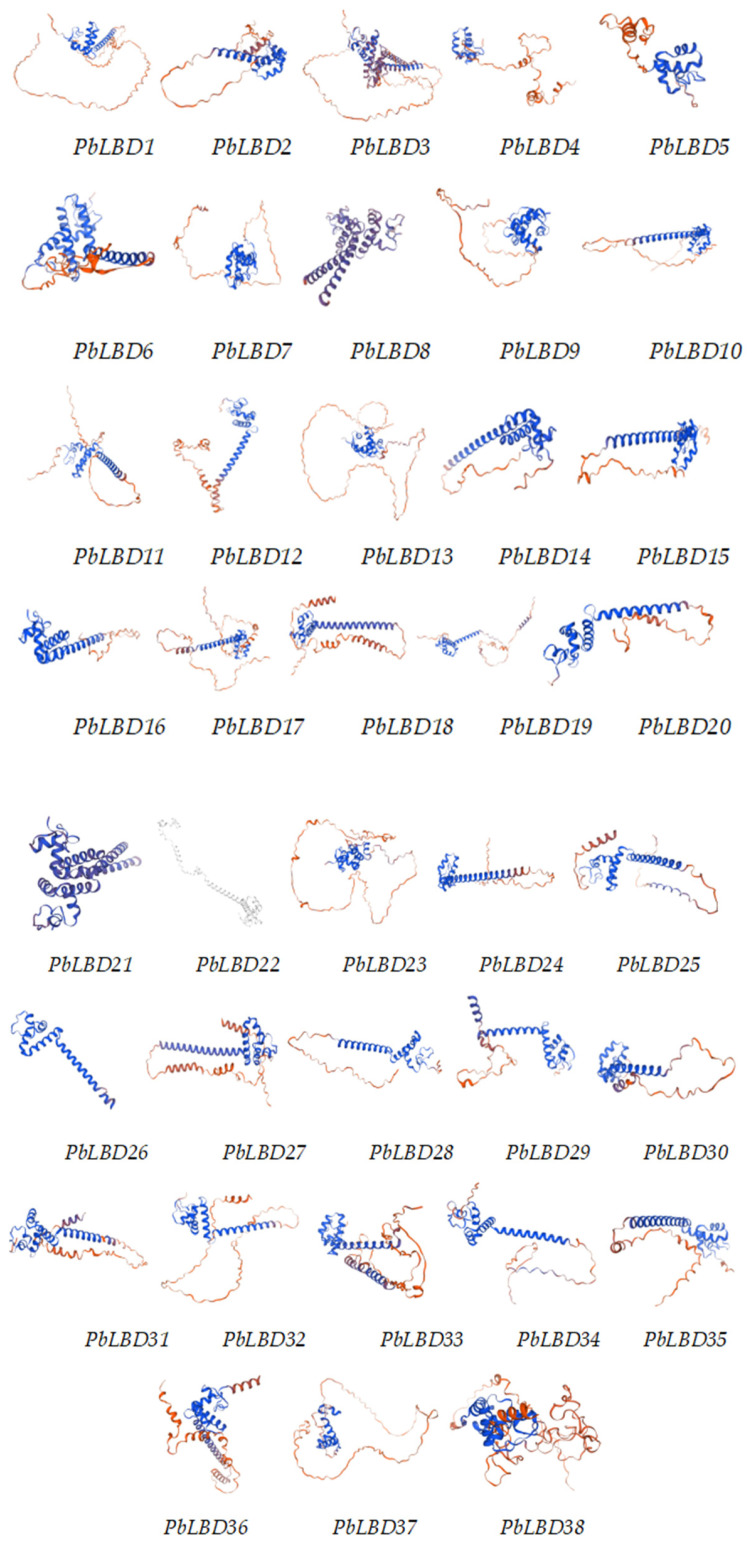
A 3D structure of PbLBD proteins based on SWISS-MODEL.

**Figure 6 ijms-24-12581-f006:**
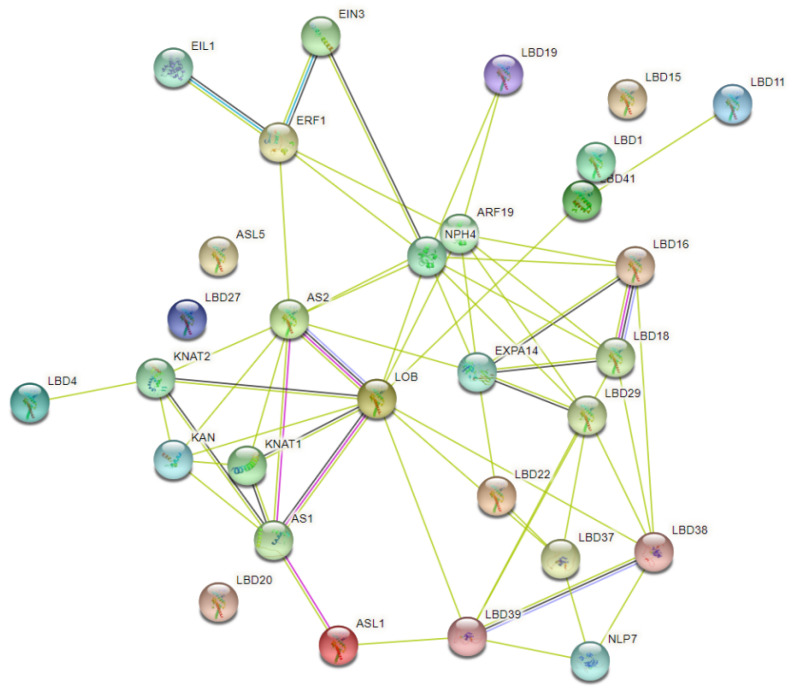
PbLBD protein interaction network. Green lines represent text mining; light blue ones mean homology; dark blue lines mean gene co-occurrence; pink ones are experimentally determined.

**Figure 7 ijms-24-12581-f007:**
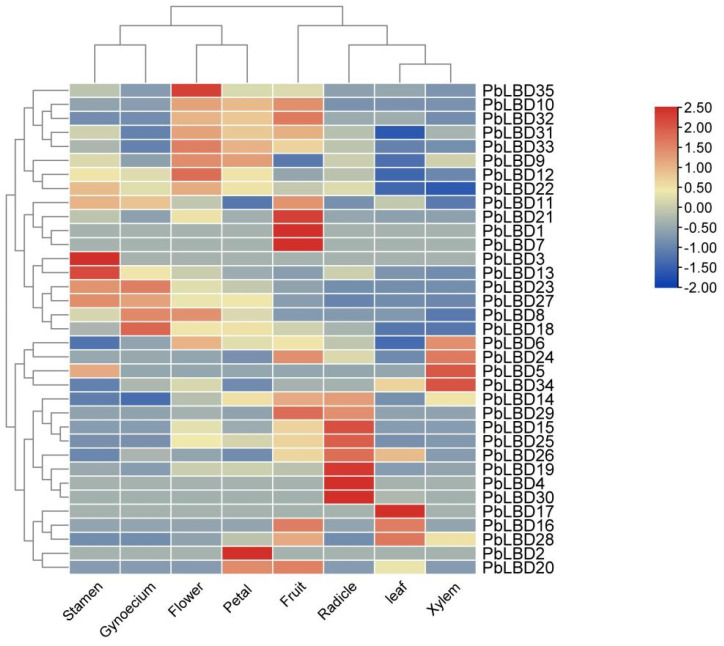
Expression of 35 *LBD* genes in various tissue of *P. bournei.* The heat map scale showed a log_2_ value from −2.00 (in dark blue) to 2.50 (in dark red) indicating a low expression level to a high expression level.

**Figure 8 ijms-24-12581-f008:**
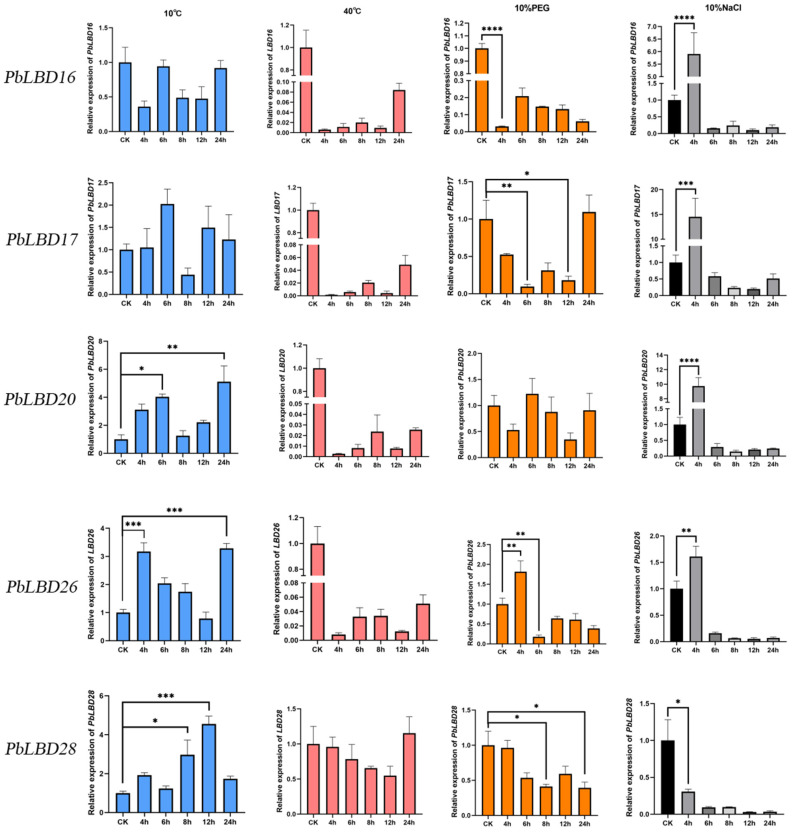
The expression of *PbLBDs* (*PbLBD16/17/20/26/28*) in leaf of *P. bournei* under cold stress (10 °C), heat stress (40 °C), salt (10%NaCl), and drought (10%PEG) stress. Different colors represent different treatments, *, **, ***, and **** showed significant differences at *p* ≤ 0.05, *p* ≤ 0.01, *p* ≤ 0.001, and *p* ≤ 0.0001, respectively.

**Figure 9 ijms-24-12581-f009:**
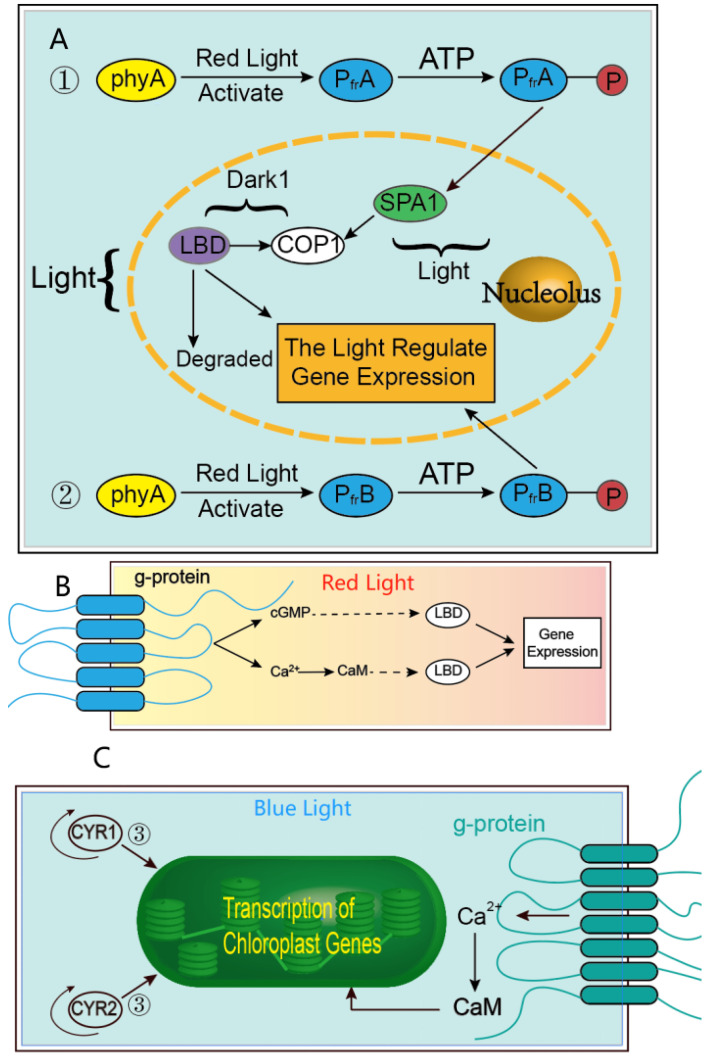
PbLBD under the red light: (**A**) passivated phytochrome A can be activated by red light and enter the nucleus through autophosphorylation across the membrane. In the presence of light, P_fr_A can activate SPA1 (suppressor of phyA 105), which is a downstream signaling component; SPA1 will further act on COP1 (constitutive photomorphogenesis 1). In the dark, COP1 may ubiquitinate the transcription factors of *P. bournei*, thereby regulating the expression of LBD genes (①) [30]. Similarly, phytochrome B (phyB) may also be involved in light-regulated gene expression, directly regulating transcription (②). (**B**) Schematic diagram of the interaction of P_fr_A and P_fr_B with G protein to regulate gene expression [31]. (**C**) Under the blue light: 1. Cryptochrome directly regulates chloroplast gene transcription (③). 2. The change of Ca^2+^ concentration induced by blue light can further promote the expression of the LBD gene in *P. bournei* [32].

**Table 1 ijms-24-12581-t001:** Physicochemical properties of the *PbLBD* genes and predicted proteins are listed.

Gene Name	Amino Acid Number	Molecular Weight (Da)	Theoretical Isoelectric Point	Total Mean Hydrophilic Value	Subcellular Localization	α Helix	Extended Strand	β Turn	Random Coil	Secondary Structure Prediction
*PbLBD1*	291	32,503.12	5.97	−0.592	Mitochondrial Matrix	39.52%	12.37%	3.44%	44.67%	Random coil
*PbLBD2*	206	22,586.44	8.18	−0.169	Chloroplast Stroma	50.00%	5.83%	1.94%	42.23%	α helix
*PbLBD3*	287	32,240.86	5.85	−0.666	Cytoplasmic Matrix	37.98%	12.89%	3.48%	45.64%	Random coil
*PbLBD4*	183	19,998.55	5.67	−0.17	Cytoplasmic Matrix	48.09%	7.10%	7.10%	37.70%	α helix
*PbLBD5*	141	16,394.41	5.92	−0.594	Peroxisome	55.32%	4.96%	4.26%	35.46%	α helix
*PbLBD6*	253	28,464.98	7.02	−0.22	Chloroplast Stroma	53.36%	7.51%	3.95%	35.18%	α helix
*PbLBD7*	207	22,321.56	6.73	−0.066	Mitochondrial Matrix	29.95%	12.08%	6.76%	51.21%	Random coil
*PbLBD8*	152	17,386.88	8.72	−0.369	Chloroplast Stroma	51.32%	2.63%	1.97%	44.08%	α helix
*PbLBD9*	202	21,241.49	7.46	0.075	Mitochondrial Matrix	24.26%	11.88%	6.93%	56.93%	Random coil
*PbLBD10*	211	23,136.05	5.32	−0.261	Chloroplast Stroma	43.60%	8.06%	1.90%	46.45%	Random coil
*PbLBD11*	256	28,282.9	9.49	−0.758	Cell Nucleus	32.03%	5.86%	3.52%	58.59%	Random coil
*PbLBD12*	245	27,093.06	9.33	−0.312	Mitochondrial Matrix	43.27%	8.57%	4.08%	44.08%	Random coil
*PbLBD13*	301	32,864.15	5.33	−0.385	Mitochondrial Matrix	30.90%	13.62%	4.32%	51.16%	Random coil
*PbLBD14*	284	32,289.83	4.94	−0.187	Mitochondrial Matrix	55.99%	8.10%	3.52%	32.39%	α helix
*PbLBD15*	200	21,392.35	6.81	−0.237	Chloroplast Stroma	50.50%	5.00%	0.50%	44.00%	α helix
*PbLBD16*	157	17,164.56	6.94	−0.166	Mitochondrial Matrix	50.32%	8.92%	0.64%	40.13%	α helix
*PbLBD17*	376	40,985.39	9.25	−0.437	Cell Nucleus	26.60%	17.29%	3.72%	52.39%	α helix
*PbLBD18*	285	33,259.55	8.08	−0.68	Endoplasmic Reticulum Membrane	54.74%	7.72%	3.16%	34.39%	α helix
*PbLBD19*	228	24,386.8	8.26	−0.178	Chloroplast Stroma	39.47%	2.63%	2.19%	55.70%	Random coil
*PbLBD20*	163	18,185.83	6.8	−0.283	Cytoplasmic Matrix	46.01%	6.75%	1.23%	46.01%	α helix/Random coil
*PbLBD21*	120	13,443.3	7.64	−0.384	Mitochondrial Matrix	64.17%	1.67%	0.83%	33.33%	α helix
*PbLBD22*	295	32,025.06	5.65	−0.26	Mitochondrial Matrix	46.10%	9.83%	4.75%	39.32%	α helix
*PbLBD23*	312	33,954.3	6.61	−0.415	Mitochondrial Matrix	28.53%	12.50%	4.81%	54.17%	Random coil
*PbLBD24*	196	21,207.35	8.13	−0.104	Chloroplast Stroma	49.49%	5.61%	1.53%	43.37%	α helix
*PbLBD25*	251	27,364	7.71	−0.277	Mitochondrial Matrix	39.84%	12.35%	2.39%	45.42%	Random coil
*PbLBD26*	165	18,082.88	8.58	−0.106	Cytoplasmic Matrix	50.91%	9.70%	1.82%	37.58%	α helix
*PbLBD27*	313	36,590.37	8.08	−0.703	Cytoplasmic Matrix	53.04%	8.31%	3.19%	35.46%	α helix
*PbLBD28*	170	18,630.24	8.59	−0.265	Cell Nucleus	42.94%	12.94%	0.59%	43.53%	Random coil
*PbLBD29*	187	20,442.35	8.29	−0.369	Mitochondrial Matrix	46.52%	5.88%	2.14%	45.45%	α helix
*PbLBD30*	172	18,557.34	8.51	0.005	Mitochondrial Matrix	49.42%	3.49%	2.33%	44.77%	α helix
*PbLBD31*	200	21,829.1	7.56	−0.152	Cytoplasmic Matrix	45.50%	9.50%	1.50%	43.50%	α helix
*PbLBD32*	232	25,559.78	5.96	−0.234	Peroxisome	39.66%	4.31%	3.88%	52.16%	Random coil
*PbLBD33*	266	29,410.15	6.42	−0.544	Mitochondrial Matrix	40.98%	6.77%	1.88%	50.38%	Random coil
*PbLBD34*	192	20,886.83	8.87	−0.231	Cell Nucleus	41.67%	10.94%	3.12%	44.27%	Random coil
*PbLBD35*	188	21,295.17	6.41	−0.314	Mitochondrial Matrix	58.51%	3.19%	0.00%	38.30%	α helix
*PbLBD36*	412	46,401.64	7.15	−0.619	Cell Nucleus	51.21%	8.74%	6.07%	33.98%	α helix
*PbLBD37*	265	28,549.46	5.02	−0.095	Mitochondrial Matrix	32.08%	12.45%	9.43%	46.04%	Random coil
*PbLBD38*	223	24,319.44	7.06	−0.371	Cytoplasmic Matrix	27.35%	13.45%	4.48%	54.71%	Random coil

**Table 2 ijms-24-12581-t002:** Information of primers.

Gene Name	Forward Primer (5′-3′)	Reverse Primer (5′-3′)
*PbLBD16*	CTGCATCTTTGCACCCTATTTC	CAGTTCCTGGAGCATCTTGT
*PbLBD17*	CAACAGCAGCAACAGAACATAG	GTATCCGTAAGGTCAAAGCTAGAG
*PbLBD20*	CAACAGGATCAGTCACCCTTAC	ACATTGCTAGAAGAGGCATAGTT
*PbLBD27*	AGCCAAGTCTCCCAATTACAG	GGAGCTGGCCAAGATAGATAAA
*PbLBD28*	GAGCCACAGAAGTTTGCTAATG	CACTCTCGCATTTGCTTCATAC

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
