# Peer review of "Genome Identification and Evolutionary Analysis of LBD Genes and Response to Environmental Factors in Phoebe bournei"

_ijms, 2023, doi:10.3390/ijms241612581_

Round 1
Reviewer 1 Report
In the present study, the authors have investigated the evolution and structure of the LBD Genes in Phoebe bournei and they also have disclosed the expression profile of these genes in response to environmental factors and light. In my opinion, the manuscript is not well written and the novelty of work is not enough for publishing in IJMS. Other comments:
- The title of the article is not the same as the title of the system.
- Sectioning of the introduction is not necessary.
- Line 100: Scientific name should be provided in italic. Please check the whole text.
- INTRODUCTION needs to improve.
- In my opinion, Figure 1 can be deleted.
- Bootstrap values should be added to figure 2.
- 371-380: gene names should be in italic format.
Author Response
Independent Review Report, Reviewer 1
In the present study, the authors have investigated the evolution and structure of the LBD Genes in Phoebe bournei and they also have disclosed the expression profile of these genes in response to environmental factors and light. In my opinion, the manuscript is not well written and the novelty of work is not enough for publishing in IJMS. Other comments:
The title of the article is not the same as the title of the system.
Response: Thanks for your revision. We apologize that we failed to change the title of the system and we will change it.
INTRODUCTION needs to improve.
Response: Sorry for the language problem, we had improved the language and added the necessary citations to make the description more clear and complete.
Sectioning of the introduction is not necessary.
Response: Thank you for your suggestion, but we are afraid that we have to have sections so that we can better illustrate transcription factors, LBD family and other significant concept based on our study and their current situation.
Line 100: Scientific name should be provided in italic. Please check the whole text.
Response: We have corrected the wrong ones.
In my opinion, Figure 1 can be deleted.
Response: Thanks for your suggestion. We have deleted the excessive one but remain another in order to explain the following content clearly.
Bootstrap values should be added to figure 2.
Response: We have added the values under the figure.
371-380: gene names should be in italic format.
Response: The format has changed as suggested.

Reviewer 2 Report
The study presented focuses on the LBD gene family in Phoebe bournei. The authors aimed to identify and analyze the LBD genes in P. bournei, characterize them, predict interactions and examine their response to different light qualities. The study used a combination of bioinformatics data, transcriptome data, and gene expression analysis. The results showed that red light had the strongest effect on the relative expression of LBD transcription factors compared to blue and white light.
Overall, the use of English should be improved across the manuscript
Introduction
In the introduction authors claim that there are several studies on LBD but they do not provide the citations
The introduction about transcription factors offers an overview, however it lacks more context on transcription regulation (TF activation)
Methods
The LBD predictions should have a better description of the domains that were mined for completeness in the candidates, and include a comparison to a closer relative with a better genome assembly/annotation
Other methods for protein prediction should have been used since they have much more improved results, like alpha fold
The methods require extensive description of the transcriptomic dataset and analysis of the gene expression patterns, including: experimental design, library preparation, platform, sequencing depth, data processing, normalization, etc.
Same for the protein interaction network. It is necessary to describe the methods more carefully and explain how the data was filtered. Since string shows multiple relational data it is important to filter the predicted interaction based on the needs of the study. Besides, it would be important to check other databased with experimentally validated data as BIOGRID
The qPCR requires explanation od what reference gene was used and why. The results from this section depend completely of that selection. qPCR is not the best method since it has a lot of artifacts. Allowing the use of transcriptome-wide measurements you could have evaluated the expression of the LBD genes and possible transcriptional targets. It is necessary to justify the qPCR selection in this case
Results
Authors should explain how the subcellular localization of the LBD proteins was determined, and discuss it. What does it mean when it is predicted to be localized to the Mitochondrial matrix or the ER? And for this family specifically, what does it mean?
The section on protein motifs, structure and interactions has a lot of potential. The authors must focus on integrating them better and extract more information from there to identify LBD top candidates for regulating the stress response
The section The Variance of LBD Gene Expression Level in Different Tissues must discuss possible sources of transcript levels based on stimuli and time. As this is not a time course, there is no certainty about the comparisons across multiple tissues. They should explain better under what conditions the measurements were taken and if there are all constant for all tissues. Having a dataset with different conditions and time course for each tissue would offer more confidence in the findings on what LBD genes have more prevalence in each tissue
Line 383 “white” instead of “while”
There are 2 figure 9 in the study and none are mentioned in the main text
Discussion
The model presented in the second figure 9 shows the chloroplast as targets for gene regulation associated with blue light. Other subcellular localizations reported for other LBD genes and the rest of the collected evidence should also be included in the model
To associate the LBD genes with the environmental response, authors do not show any evidence in P bournei. They only measure the expression response after the stimuli. This does not prove them implications in the plant response, and as they did not do transcriptome-wide analysis they could not do any models like GRNs or co-expression networks to see if the LBD genes were proposed as top regulators in the transcriptional response. They did not do any functional analysis with he top candidates that they found to see if their silencing/depletion caused changes in the overall response. Because of that the study cannot link LBD genes as regulators of the response.
I suggest that the authors focus on what they can claim from the obtained data and discuss other sources of variation/experimental design that can explain what they observed
Author Response
Independent Review Report, Reviewer 2
The study presented focuses on the LBD gene family in Phoebe bournei. The authors aimed to identify and analyze the LBD genes in P. bournei, characterize them, predict interactions and examine their response to different light qualities. The study used a combination of bioinformatics data, transcriptome data, and gene expression analysis. The results showed that red light had the strongest effect on the relative expression of LBD transcription factors compared to blue and white light.
Overall, the use of English should be improved across the manuscript
Response: Thank you for your comments. We have improved and edited the text further, based on helpful comments.
Introduction
In the introduction authors claim that there are several studies on LBD but they do not provide the citations
The introduction about transcription factors offers an overview, however it lacks more context on transcription regulation (TF activation)
Response: We have added literature on the introduction of LBD and an explanation of the importance of TF.
Methods
The LBD predictions should have a better description of the domains that were mined for completeness in the candidates, and include a comparison to a closer relative with a better genome assembly/annotation
Other methods for protein prediction should have been used since they have much more improved results, like alpha fold
Response: We have made modifications to the content in section 4.2.1 and increased the validation standards of the SMART and InterPro databases to confirm that all of the candidate genes have the LBD conserved domain. To confirm their genetic relationships, we constructed a phylogenetic tree in section 4.2.3 using the LBD gene sequences of the model species Arabidopsis and the LBD candidate genes of P. bournei. Also to improve the results, the proteins prediction was made together with the table of genes. We use the PRABI database to further predict the secondary domain prediction values of LBD protein in P. bournei, making the data more complete. L462-476
The methods require extensive description of the transcriptomic dataset and analysis of the gene expression patterns, including: experimental design, library preparation, platform, sequencing depth, data processing, normalization, etc.
Response: Thanks for your advice. We used the previously proven reliable transcriptome data in the laboratory to screen the differential expression data of LBD gene family members in different tissues of P. bournei through local blast comparison.
Same for the protein interaction network. It is necessary to describe the methods more carefully and explain how the data was filtered. Since string shows multiple relational data it is important to filter the predicted interaction based on the needs of the study. Besides, it would be important to check other databased with experimentally validated data as BIOGRID
Response: We have added relevant screening details using String database. We apologize that we did not use the biogrid database as we are not familiar with this database. Secondly, by online learning, we have not found the results of the related interaction of LBD gene family in the template of Arabidopsis in this database for the time being. Also the fitting results of PbLBD protein to corresponding proteins in Arabidopsis was uploaded in supplementary materials. L484-487
The qPCR requires explanation od what reference gene was used and why. The results from this section depend completely of that selection. qPCR is not the best method since it has a lot of artifacts. Allowing the use of transcriptome-wide measurements you could have evaluated the expression of the LBD genes and possible transcriptional targets. It is necessary to justify the qPCR selection in this case
Response: The internal reference gene we selected is based on the stable expression gene PbEF1 obtained from previous studies (Zhang et al., 2018). The reason for choosing these five LBD candidate genes that are correlated with temperature is because we believe that the P. bournei is a mid-subtropical tree species, with the warm and humid habitat that is widely distributed in Southern China (Danmei, Jiping and Tao, 2022). We hope to understand the impact of temperature changes on the genes of P. bournei, in order to provide direction for future research on expanding the cultivation of this tree species. The reason for choosing qPCR is that by comparing the fluorescence data of these genes under different treatments with the standard internal reference gene data, we can directly see whether the expression of these genes under different treatments is correct. We have also considered the artifacts of qPCR. Therefore, our experimental data is based on three biological replicates for each different treatment, and three mechanical replicates for each biological replicate sample. L551-553
Results
Authors should explain how the subcellular localization of the LBD proteins was determined, and discuss it. What does it mean when it is predicted to be localized to the Mitochondrial matrix or the ER? And for this family specifically, what does it mean?
Response: We have corrected the figure in a statistical method with the statistical significance of the differences in gene expression in the Figure 8. The subcellular localizations of these candidate genes were predicted by Wolf PSORT tool and we have modified the article and added the following in 3.1 content according to the subcellular structure prediction, most of the LBD gene families are expressed localized on mitochondria versus chloroplasts, the same subcellular localizations are representative for plants, which correspond to the functions of the conserved domains of this gene family ie in plant lateral organs such as leaf The morphogenesis of flowers and others coincides with an important role in growth and development (Shuai, Reynaga and Springer, 2002). L330-340
The section on protein motifs, structure and interactions has a lot of potential. The authors must focus on integrating them better and extract more information from there to identify LBD top candidates for regulating the stress response
The section The Variance of LBD Gene Expression Level in Different Tissues must discuss possible sources of transcript levels based on stimuli and time. As this is not a time course, there is no certainty about the comparisons across multiple tissues. They should explain better under what conditions the measurements were taken and if there are all constant for all tissues. Having a dataset with different conditions and time course for each tissue would offer more confidence in the findings on what LBD genes have more prevalence in each tissue
Response: What we based on is that under various kinds of treatment how the genes of our samples in the different tissues express. Instead of comparing across multiple tissues, we hope to understand the expression of different LBD candidate genes in the same tissue. This allows us to determine which candidate genes have more significant expression, which usually indicates better regulatory function. This provides clearer evidence for subsequent studies. Since in previous studies we have found that the LBD gene family has a definite function of playing an important role in the growth and development of lateral organs, such as leaves and flowers, we chose PbLBD16/17/20/26/28 in leaves and flowers that have higher expression level than other genes. Additionally, as P. bournei species mainly live in warmer regions in the middle subtropical climate, we want to know whether non-biotic stress such as temperature stress has an impact using q-PCR to validate. L523
Line 383 “white” instead of “while”
Response: Sorry for carelessness. We have corrected it.
There are 2 figure 9 in the study and none are mentioned in the main text
Response: We mistook to arrange the figures and we have re-arranged them again so that they are consistent with the content.
Discussion
The model presented in the second figure 9 shows the chloroplast as targets for gene regulation associated with blue light. Other subcellular localizations reported for other LBD genes and the rest of the collected evidence should also be included in the model
To associate the LBD genes with the environmental response, authors do not show any evidence in P bournei. They only measure the expression response after the stimuli. This does not prove them implications in the plant response, and as they did not do transcriptome-wide analysis they could not do any models like GRNs or co-expression networks to see if the LBD genes were proposed as top regulators in the transcriptional response. They did not do any functional analysis with he top candidates that they found to see if their silencing/depletion caused changes in the overall response. Because of that the study cannot link LBD genes as regulators of the response.
I suggest that the authors focus on what they can claim from the obtained data and discuss other sources of variation/experimental design that can explain what they observed
Response: As mentioned above, we want to understand the expression of different LBD candidate genes in the same tissue in order to better identify which candidate genes have more obvious expression. Higher levels of expression usually represent better regulatory functions. We validated the differences in their expression levels under different long-term stress conditions of temperature, salt, and drought using qRT-PCR analysis. It can also be seen that we can predict the response patterns of the selected genes to stress. Thank you for your comments. We will consider conducting the transcriptome analysis for further research.

Reviewer 3 Report
Comments to the Author(s):
The manuscript “Genome Identification and Evolutionary Analysis of LBD Genes and Response to Environmental Factors in Phoebe bournei” (Ms No. ijms-2343257) by Yiming Ma and colleagues presents a global analysis of LBD transcription factor gene family in Phoebe bournei genome. The study mainly includes genome-wide survey and bioinformatics analyses of LBD genes in P. bournei genome, as well as development and stress expression analyses of LBD genes. This type of study falls within the scope of IJMS. The reason to perform this study is convictive. The material, methods, and protocols were well applied for most of the study. However, though the manuscript contains some original data, it was not well organized and wrote; and the main results are not innovative and lack deep analyses. So, I can not recommend this manuscript for publication. Some major comments are supplied below.
Major comments:
1. The “Abstract” part should be need to be modified to clearly summary the most important findings in the manuscript.
2. The “Introduction” part was long and was not well organized, which needs to be revised significantly. I suggested the authors to focus on the contents related to the theme of the ms. For instance, the contents in line 53-74 are too long which are not very close to the theme of the ms. Moreover, the subheadings in this part are unnecessary.
3. The main contents and results of the ms are too simple and are very descriptive. Most of the parts lack deep analysis. Moreover, (1) “Table 1” should be simplified, and some common information could be supplied as Supplemental data. (2) The classification of candidate LBD proteins should be checked carefully, as the members of Class II were not clustered in the some branch in the NJ tree in “Figure 2”? I suggest the authors to check the parameters in constructing the NJ tree, including the missing/gap treatment and substitution model. (3) The contents in “Figure 3” are not related to “Gene Structure”? They are “Protein Structure”? And the “left” and “right” structure diagrams should be integrated into one as they commonly indicated the protein structure feature of this gene family. Moreover, the authors should explain why the members in Class II have different protein structure feature. (4) The information of the conserved protein domains in this gene family should be marked in “Figure 4” as mentioned in the text. (5). “Figure 5” was poorly edited with little information, which needs figure legend. (6). The expressions of LBD genes need deep analysis, and it is better to compare the expression of homologs in the same class, as well as between different classes.
4. The “Discussion” part needs to be revised significantly, which was not well organized. The authors need to cite more references to support your statements. Moreover, the contents related to “Figure 9” were unconvincing, as this study just did expression anlayses of a few candidates. The function and mechanism of the LBD genes in P. bournei need to be verified by experiments.
5. The “Conclusion” part is long which is not innovative or is not highlighted enough.
6. There are many inaccuracies statements and even grammar mistakes in the full text which should be corrected as well.
Author Response
Independent Review Report, Reviewer 3
The manuscript “Genome Identification and Evolutionary Analysis of LBD Genes and Response to Environmental Factors in Phoebe bournei” (Ms No. ijms-2343257) by Yiming Ma and colleagues presents a global analysis of LBD transcription factor gene family in Phoebe bournei genome. The study mainly includes genome-wide survey and bioinformatics analyses of LBD genes in P. bournei genome, as well as development and stress expression analyses of LBD genes. This type of study falls within the scope of IJMS. The reason to perform this study is convictive. The material, methods, and protocols were well applied for most of the study. However, though the manuscript contains some original data, it was not well organized and wrote; and the main results are not innovative and lack deep analyses. So, I can not recommend this manuscript for publication. Some major comments are supplied below.
Response: Dear reviewer, thank you for your recognition of our research. According to your comments, we have revised the relevant content of the paper again. Thank you for your valuable comments.
- The “Abstract” part should be need to be modified to clearly summary the most important findings in the manuscript.
Response: Thank you for your comments. We delete the redundant content in the abstract, and further improve the content of the abstract, so that the abstract can better reflect the content of the article.
- The “Introduction” part was long and was not well organized, which needs to be revised significantly. I suggested the authors to focus on the contents related to the theme of the ms. For instance, the contents in line 53-74 are too long which are not very close to the theme of the ms. Moreover, the subheadings in this part are unnecessary.
Response: Thanks for your comments, we have discussed and rewritten the content of the introduction in order to express our content more logically. L42-67
“Table 1” should be simplified, and some common information could be supplied as Supplemental data.
Response: We built TABLE1 by learning from IJMS's previous submissions. IThanks for your comments, we considered eliminating the redundant parts and adding content construction to improve the article. We made the correspondence between the original gene id of P. bournei corresponding to PBlbd and PBlbd into the attached table, and added ORF to the attached table *bp (open reading Frame) data.
The classification of candidate LBD proteins should be checked carefully, as the members of Class II were not clustered in the some branch in the NJ tree in “Figure 2”? I suggest the authors to check the parameters in constructing the NJ tree, including the missing/gap treatment and substitution model.
Response: Sorry for the error in the description earlier. We used MUSCLE software to perform protein sequence alignment, and IQTREE software to set local comparison based on maximum likelihood method and construct an evolutionary tree with 1000 repetitions as the parameter. Then MEGA software was used to visualize the phylogenetic tree. The construction of the phylogenetic tree of P. bourneiand Arabidopsis has no problem in cluster analysis, but it may have code errors when beautifying with evoview. The has been reworked and the bootstrap confidence level classification identifier has been added. L165-171
The contents in “Figure 3” are not related to “Gene Structure”? They are “Protein Structure”? And the “left” and “right” structure diagrams should be integrated into one as they commonly indicated the protein structure feature of this gene family. Moreover, the authors should explain why the members in Class II have different protein structure feature.
Response: Thank you for your comments. There was an error in writing the title. It has been fixed. Figure 3 shows the conserved protein structure diagram of this gene family, which mainly includes the phylogenetic tree on the left, the motif conserved motif on the left, and the protein domain conserved domain on the right. The left and right are actually a graph. Since motif and domain are not the same concept, we did not merge at the lower 5-3 ends. They're alone and separated. Previous studies have confirmed that a class2 is divided into LBD gene family because of the existence of incomplete LOB conserved domain, and we will emphasize the reason of classification in this paper. Subgroup 2 is divided because of the existence of discontinuous LOB conserved domain (verified by interpro). The whole Class2 they are short in length and there are missing motif motif or broken part, which we have added in this part. L185
The information of the conserved protein domains in this gene family should be marked in “Figure 4” as mentioned in the text.
Response: The LOB conserved domain is mainly composed of three fragments, among which the two conserved motifs are the main conserved part. According to previous studies and investigations, the LBD gene family is also divided into two subgroups according to whether the conserved motifs on the right in Figure 4 are complete or not, so we chose this composition. Thanks for your reminding, we will explain it in the figure annotation in the paper.
“Figure 5” was poorly edited with little information, which needs figure legend.
Response: We re-edited FIGURE5 and used the homology modeling of SWISS-model to predict and supplement the protein level 3 structure of other LBD genes based on GMQE value, and at the same time added relevant content in the paper. L229-231, 468-469
The expressions of LBD genes need deep analysis, and it is better to compare the expression of homologs in the same class, as well as between different classes.
Response: Due to the limitations of conditions, based on the characteristics of the LBD gene family, which mainly plays a role in the lateral organs, such as leaves and fruits, we selected these 5 genes with high expression in the comparison between leaves and fruits according to the transcriptome data. Through the expression analysis of these five significantly expressed genes, we could see the expression of these genes under different conditions. The expression of these significant genes was used to observe the expression of LBD gene family in P. bournei species.
The “Discussion” part needs to be revised significantly, which was not well organized. The authors need to cite more references to support your statements. Moreover, the contents related to “Figure 9” were unconvincing, as this study just did expression anlayses of a few candidates. The function and mechanism of the LBD genes in P. bournei need to be verified by experiments.
Response: Thanks for your comments, we revised the discussion part and sought relevant literature to verify our experimental results. Due to the limited experimental conditions, we can only select the combination of the functional characteristics of LBD gene family for lateral organs and the differential expression data of RNA-seq transcriptome to select the most significant genes for testing, and through these conditions, we can simulate the characteristics of LBD gene family in Fujian P. bournei species so as to provide references for subsequent studies. L332-335, 339-348
The “Conclusion” part is long which is not innovative or is not highlighted enough.
Response: Thank you for your valuable comments. We have deleted unnecessary parts of the conclusion and revised the content to highlight our research focus in the conclusion.
There are many inaccuracies statements and even grammar mistakes in the full text which should be corrected as well.
Response: Thank you for your correction. We have sought out teachers in English-speaking countries to revise our article
Reviewer 4 Report
In the manuscript entitled "Genome Identification and Evolutionary Analysis of LBD Genes and Response to Environmental Factors in Phoebe bournei" the authors investigated the physiological properties, gene structures, phylogenetic relationships, and expression pattern of 38 LBD genes in P. bournei tree. The research work is well-planned and performed, the methods were described in detail. In general, the manuscript is written in a good manner and order.
Further comments:
- -Line #116 write only Figure 1
- Table 1.: If possible, reduce the letter size a little bit, or at least the spacing between the lines to make the table less voluminous. Also, better to mention the units in the heading row of the table (in the case of molecular weight).
- Figure 6.: Mention the meaning of different line colours in the figure caption.
- The comments from Xue-Rong Zhou are left in the text. Please, remove those and act accordingly to the suggestions.
Author Response
Independent Review Report, Reviewer 4
In the manuscript entitled "Genome Identification and Evolutionary Analysis of LBD Genes and Response to Environmental Factors in Phoebe bournei" the authors investigated the physiological properties, gene structures, phylogenetic relationships, and expression pattern of 38 LBD genes in P. bournei tree. The research work is well-planned and performed, the methods were described in detail. In general, the manuscript is written in a good manner and order.
Response: We have further modified the article according to the teacher and your comments, thank you for your recognition.
-Line #116 write only Figure 1
-Table 1.: If possible, reduce the letter size a little bit, or at least the spacing between the lines to make the table less voluminous. Also, better to mention the units in the heading row of the table (in the case of molecular weight).
Response: We have made a new schedule, and added and improved the content, which can better make the table clear, and also better express the content of the paper.
- Figure 6.: Mention the meaning of different line colours in the figure caption.
-The comments from Xue-Rong Zhou are left in the text. Please, remove those and act accordingly to the suggestions.
Response: Thanks for your revision. We have changed the above ones as suggestions.
Round 2
Reviewer 1 Report
I had previously rejected this manuscript because its innovation and content were not at the level of the IJMS. In the revised version, the flaws are still present. Introduction is written as a review article. Figure 2 is not acceptable and in addition, more evolution analyses such as prediction of duplicated gene, Ka/Ks, etc. should be added to manuscript. The discussion is not well written and the results are not well interpreted.
- Lines 28-29: “The LBD gene structure is not complex” could be deleted.
- Line 35: PbLBDs be provided in italic format. Please use italic for gene name NOT gene family name.
- “Protein sequences” can be removed from Keywords.
- Based on figure 2, ATLBD 12, 29, 36, and 40 should be grouped in Ic NOT Ib. Also, bootstrap values are not recognizable on the figure.
- Line 387: Correct the scientific name of tomato “(Lycopersiconesculentum)”
- Neighbor-joining method is not recommended for phylogeny analysis. ML method is used.
Author Response
Independent Review Report, Reviewer 1
I had previously rejected this manuscript because its innovation and content were not at the level of the IJMS. In the revised version, the flaws are still present. Introduction is written as a review article. Figure 2 is not acceptable and in addition, more evolution analyses such as prediction of duplicated gene, Ka/Ks, etc. should be added to manuscript. The discussion is not well written and the results are not well interpreted.
Response: Thanks to the reviewers for their valuable comments, we reviewed the content of the paper again and made great efforts to modify the parts that we could modify and supplement according to existing conditions. We re-wrote the introduction content, and reconstructed the phylogenetic tree in FIG. 2 based on the homologous protein sequence of Arabidopsis related genes with an e value of 10-5. In the discussion and results, the writing was further strengthened to better express our research conclusions.
- Lines 28-29: “The LBD gene structure is not complex” could be deleted.
- Line 35: PbLBDs be provided in italic format. Please use italic for gene name NOT gene family name.
- “Protein sequences” can be removed from Keywords.
Response: Thank you for your valuable advice, we have corrected the above errors, thank you for your reminder.
Based on figure 2, ATLBD 12, 29, 36, and 40 should be grouped in Ic NOT Ib. Also, bootstrap values are not recognizable on the figure.
Response: Thanks to your comments, we have re-reviewed the diagram and revised it using maximum likelihood method and added confidence guide markers based on the sequence results of Arabidopsis homologous proteins.
Neighbor-joining method is not recommended for phylogeny analysis. ML method is used.
Response: Thank you for your suggestion. We re-use the maximum likelihood method to make the phylogenetic tree
Reviewer 2 Report
The manuscript has been extensively improved. The authors still must add more details to section 4.2.5: this section requires extensive description of the transcriptomic dataset and analysis of the gene expression patterns, including: experimental design, library preparation, platform, sequencing depth, data processing, normalization, etc.
Author Response
Independent Review Report, Reviewer 2
The manuscript has been extensively improved. The authors still must add more details to section 4.2.5: this section requires extensive description of the transcriptomic dataset and analysis of the gene expression patterns, including: experimental design, library preparation, platform, sequencing depth, data processing, normalization, etc.
Response: Thank you for your comments. We have added relevant information and reference according to your suggestion. L509-531
Round 3
Reviewer 1 Report
The current version is improved.
Author Response
Thank you for reviewing our paper, and thank you for giving us this opportunity, we know that we still have many shortcomings, we will continue to work hard in the future, and I wish you good health and success in your career。